# Prediction of Cell-Penetrating Peptides Using a Novel HSIC-Based Multiview TSK Fuzzy System

**Peng Liu** [1], **Shulin Zhao** [1,2], **Quan Zou** [1,2] and **Yijie Ding** [2,*]

[1] Institute of Fundamental and Frontier Sciences, University of Electronic Science and Technology of China, Chengdu 611731, China; 202021210231@std.uestc.edu.cn (P.L.); 202021210230@std.uestc.edu.cn (S.Z.); zouquan@nclab.net (Q.Z.)

[2] Institute of Yangtze Delta Region (Quzhou), University of Electronic Science and Technology of China, Quzhou 324000, China

[*] Correspondence: wuxi_dyj@163.com

**Abstract:** Cell-penetrating peptides (CPPs) are short peptides that can carry cargo into cells. CPPs are widely utilized due to their powerful loading capacity and transduction efficiency. Identifying CPPs is the basis for studying their functions and mechanisms; however, experimental methods to identify CPPs are expensive and time-consuming. Recently, CPP predictors based on machine learning methods have become a research hotspot. Although considerable progress has been made, some challenges remain unresolved. First, most predictors employ a variety of feature descriptors to transform an original sequence into multiview data; however, extant methods ignore the relationships between different views, limiting further performance improvement. Second, most machine learning models are actually black boxes and cannot offer insightful advice. In this paper, a novel Hilbert–Schmidt independence criterion (HSIC)-based multiview TSK fuzzy system is proposed. Compared with other machine learning methods, TSK fuzzy systems have better interpretability, and the introduction of multiview mechanisms provides comprehensive insight into the intrinsic laws of the data. HSIC is utilized here to measure the independence and enhance the complementarity between different views. Notably, the proposed method attained prediction accuracy results of 92.2% and 96.2% for the training and independent test sets, respectively. The empirical results show that our promising approach features greater recognition performance than the state-of-the-art method.

**Keywords:** cell-penetrating peptides; machine learning; TSK fuzzy system; multiview learning; HSIC

## 1. Introduction

Traditional therapeutic drugs are greatly limited due to the complexity of the human immune system and the selective penetration of the cell membrane. As such, many diseases require treatment at the molecular level. We expect to deliver drugs directly to target cells while minimizing the impact on cells and avoiding permanent damage. Cell-penetrating peptides (CPPs) can be used to complete this task. CPPs are a class of short peptides with a length between 5–50 amino acid residues [1] that can carry DNA, protein, and other biomolecules into cells and will not cause irreparable damage to cells when the concentration of CPPs is low. CPPs are widely utilized due to their powerful loading capacity and transduction efficiency. Therefore, the correct identification of CPPs is of great significance. Unfortunately, the traditional experimental approach is time-consuming and costly to predict CPPs, and the prediction accuracy is not satisfactory.

In recent years, machine learning-based methods have been widely applied [2,3] to predict CPPs. These methods have two main steps, namely, (1) selecting a suitable feature extraction method to transform the original sequence into vector form. In this process, to reduce information loss, a variety of descriptors are often adopted to convert the sequence into multiview data. The second step is to (2) build a learning model and utilize the features obtained in the above step as input to train the model. Such machine

learning-based predictors have evolved rapidly in the past few years. CellPDD, proposed by Gautam et al. [4], adopts several feature representation methods, such as the amino acid composition, dipeptide composition, and binary spectroscopy. Diener et al. [5] improved the prediction performance by utilizing the amino acid frequency and physicochemical property features. Wei et al. constructed a high-quality dataset, CPP924, and presented two effective predictors: SkipCPP-Pred [6] and CPPred-RF [7]. An adaptive skip dipeptide composition descriptor and random forest algorithm were employed. The TargetCPP proposed by Arif et al. [8] adopted split amino acid composition and composite protein sequence representation, covering multiview information, and the gradient boost decision tree algorithm was employed to improve the prediction performance. Fu et al. [9] built a predictor named StackCPPred based on the residue pairwise energy matrix and employed support vector machine recursive feature elimination and correlation bias reduction to improve the identification ability. In addition to these predictors for CPPs, some methods have been used to predict other therapeutic peptides. PEPred, proposed by Wei et al. [10], and PPTPP, proposed by Zhang et al. [11], can be used to predict eight therapeutic peptides: AAP, ABP, ACP, AIP, AVP, CPP, QSP, and SBP. ITP-Pred, proposed by Cai et al. [12], can predict both CPP and QSP. These prediction methods improve the ability to discriminate CPPs and lay the foundation for the wide application of CPPs.

A fuzzy system is a rule-based system that implements knowledge representation via fuzzy logic and inference. The core of a fuzzy system is a knowledge base composed of IF-THEN rules. In this paper, the Takagi–Sugeno–Kang (TSK) fuzzy system [13,14] was adopted due to its excellent interpretability and data-driven learning ability [15–17].

In multiview learning, each view can benefit from knowledge from other views, which is the complementarity principle. In addition, some studies [18] have noted that the independence of different views can serve as a beneficial complement to multiview learning. In this paper, the Hilbert–Schmidt independence criterion (HSIC) [19] is employed to measure the independence of different views and realize the idea of the complementarity principle.

Although there are many methods available to predict CPPs, some critical questions remain unanswered. These problems include the following: (1) many proposed predictors adopt multiple feature descriptors, but they simply splice each feature vector and directly input the hybrid feature into the prediction model. The disadvantage of doing so is that the interaction from different views and the statistical characteristics of the data are ignored. The predictive performance is also compromised as a result. (2) Most machine learning models are actually black boxes; however, a fuzzy system with a knowledge base based on fuzzy rules has good interpretability and can provide insightful suggestions to study the underlying rules.

To solve the above problems, a CPP predictor based on a multiview TSK fuzzy system is proposed, and the main workflow of the process is shown in Figure 1. First, two feature descriptors were employed, namely, soft symmetric alignment and pseudo-amino acid composition. Then, the correlation-based feature selection algorithm was adopted to remove redundant features and noise. The resulting feature subset was input into the multiview TSK fuzzy system. Finally, the multiview decision result was obtained.

The main contributions of this study are as follows: (1) We introduce a multiview TSK fuzzy system based on HSIC. Compared with other machine learning methods, TSK fuzzy systems have advantages in interpretability. The introduction of multiview mechanisms allows for comprehensive insight into the intrinsic laws of the data. HSIC was utilized to measure the independence and enhance the complementarity between different views. (2) The proposed method is competitive or better than the state-of-the-art CPP predictors. The empirical results show that our method has broad application prospects.

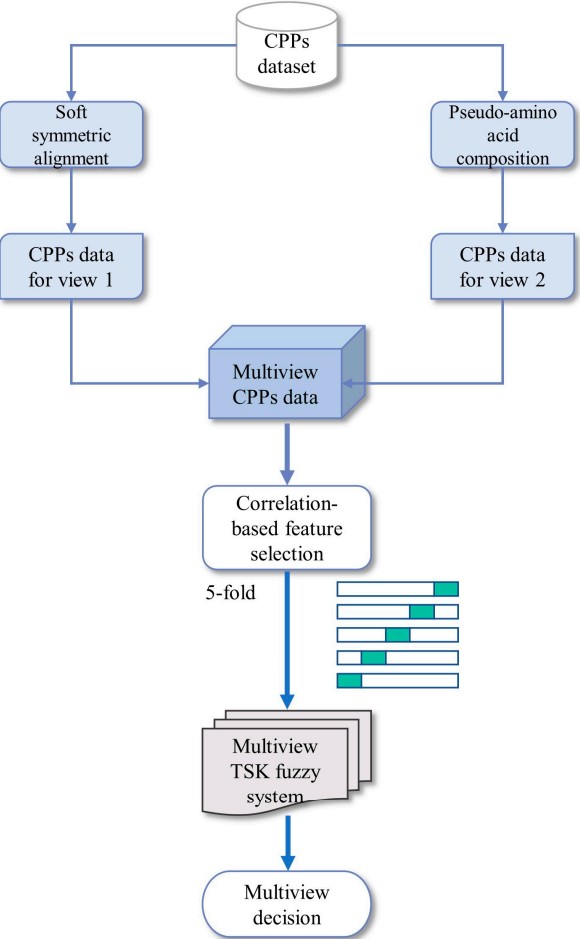

**Figure 1.** Workflow of the proposed method.

## 2. Materials and Methods

### 2.1. Data Collection

In this paper, the CPP740 [7] dataset was adopted. CPP740 contains 370 CPPs and the same amount of non-CPPs. In addition, we also employed an independent test set to validate the performance of our method. The dataset contains 92 positive samples and the same number of negative samples.

### 2.2. Feature Extraction

#### 2.2.1. Pseudo-Amino Acid Composition

During the process of converting biological sequences into vectors, it is inevitable that some information will be lost. To maximally retain the information of the original sequence, a variety of feature representation methods have been proposed. Among them, the pseudo-amino acid composition (Pse-AAC), proposed by Chou et al. [20], has been widely used in various fields of bioinformatics. Pse-AAC incorporates contiguous local sequence-order information and global sequence-order information into the feature vector. After application of this method, the sequence is represented as a 50-dimensional feature vector.

#### 2.2.2. Soft Symmetric Alignment

The soft symmetric alignment (SSA) feature was adopted by Lv et al. [21] to predict anticancer peptides. It is a deep representation learning feature extraction method. It trains a three-layer stacked BiLSTM encoder that converts the sequence into a matrix $\mathbf{R}^{L \times 121}$, where L is the length of the peptide. Then, the similarity loss function is used to optimize the model parameters through backpropagation, and the sequence is transformed into a 121-D feature vector.

### 2.3. Feature Selection

To improve the computational efficiency, eliminate redundant features, avoid overfitting problems and improve the generalization ability of the model, it is necessary to adopt a suitable feature selection method. In this paper, we employed the correlation-based feature selection algorithm (CFS) [22], which does not rank individual features but searches for the optimal subset of features. A feature subset is considered valuable if its features are highly correlated with the labels and its redundancy is low. The greedy algorithm was used to search for feature subsets, and a subset containing 37 features was selected, including 15 SSA features and 22 Pse-AAC features.

### 2.4. TSK Fuzzy System

The TSK fuzzy system is a classic fuzzy model. Its input and output are nonfuzzy values, and it is highly flexible and interpretable. Therefore, we chose the TSK fuzzy system as the basic model. A fuzzy rule in TSK can be defined as follows:

$$
\begin{aligned}
&\mathrm{R}^k : \text{IF } x_1 \text{ is } A_1^k \wedge x_2 \text{is } A_2^k \wedge \ldots \wedge x_D \text{is } A_D^k, \\
&\text{THEN } y^k(\mathbf{x}) = p_0^k + p_1^k x_1 + p_2^k x_2 \ldots + p_D^k x_D, \ k = 1, 2, \ldots, K
\end{aligned}
\tag{1}
$$

The above TSK fuzzy system consists of K rules, and the input vector $\mathbf{x} = [x_1, x_2, \ldots, x_D]^{\mathrm{T}}$. $A_d^k$ is a fuzzy set corresponding to the dth feature of the kth rule, $y^k$ is the output of the kth rule, and $p_d^k$ is the parameter. The membership function of the fuzzy set $A_d^k$ is commonly represented by a Gaussian function:

$$
\mu_{A_d^k}(x_d) = \exp\left( -\frac{\left(x_d - c_d^k\right)^2}{2\sigma_d^k} \right)
\tag{2}
$$

where $c_d^k$ denotes the center and $\sigma_d^k$ denotes the variance. In this paper, the fuzzy c-means (FCM) algorithm is employed to calculate $c_d^k$ and $\sigma_d^k$.

$$
c_d^k = \frac{\sum_{i=1}^{N} u_{ik} x_{id}}{\sum_{i=1}^{N} u_{ik}}
\tag{3}
$$

$$
\sigma_d^k = \frac{h \sum_{i=1}^{N} u_{ik} \left(x_{id} - c_d^k\right)^2}{\sum_{i=1}^{N} u_{ik}}
\tag{4}
$$

The output of the TSK fuzzy system is the combination of the results of each rule, which can be expressed as:

$$
\mathrm{y}(\mathbf{x}) = \frac{\sum_{k=1}^{K} \mu^k(\mathbf{x}) y^k(\mathbf{x})}{\sum_{k=1}^{K} \mu^k(\mathbf{x})} = \sum_{k=1}^{K} \widetilde{\mu}^k(\mathbf{x}) y^k(\mathbf{x})
\tag{5}
$$

where

$$
\mu^k(\mathbf{x}) = \prod_{d=1}^{D} \mu_{A_d^k}(x_d)
\tag{6}
$$

and

$$
\widetilde{\mu}^k(\mathbf{x}) = \frac{\mu^k(\mathbf{x})}{\sum_{k=1}^{K} \mu^k(\mathbf{x})}
\tag{7}
$$

For the input vector $\mathbf{x}$, let

$$
\mathbf{x_e} = \left(1, \mathbf{x}^{\mathrm{T}}\right)^{\mathrm{T}}
\tag{8}
$$

$$
\overset{\sim}{\mathbf{x}}^k = \widetilde{\mu}^k(\mathbf{x}) \mathbf{x_e}
\tag{9}
$$

$$\mathbf{x}_g = \left( \left( \tilde{\mathbf{x}}^1 \right)^{\mathrm{T}}, \left( \tilde{\mathbf{x}}^k \right)^{\mathrm{T}}, \ldots, \left( \tilde{\mathbf{x}}^k \right)^{\mathrm{T}} \right)^{\mathrm{T}} \tag{10}$$

then

$$\mathbf{p}^k = \left( p_0^k, p_1^k, \ldots, p_D^k \right)^{\mathrm{T}} \tag{11}$$

$$\mathbf{p}_g = \left( \left( \mathbf{p}^1 \right)^{\mathrm{T}}, \left( \mathbf{p}^2 \right)^{\mathrm{T}}, \ldots, \left( \mathbf{p}^K \right)^{\mathrm{T}} \right)^{\mathrm{T}} \tag{12}$$

$$\mathrm{y}(\mathbf{x}) = \mathbf{p}_g^{\mathrm{T}} \mathbf{x}_g \tag{13}$$

According to the above transformation, the TSK fuzzy system is transformed into a linear model. We employed the method of Deng et al. [23] to solve model coefficients. The objective function is as follows:

$$\min_{\mathbf{P}_g} J_{\mathrm{TSK}} \left( \mathbf{p}_{g,c} \right) = \frac{1}{2} \sum_{c=1}^{C} \mathbf{p}_{g,c}^{\mathrm{T}} \mathbf{p}_{g,c} + \frac{\lambda_{\mathbf{p}_g}}{2} \sum_{c=1}^{C} \sum_{i=1}^{N} \parallel y_{ic} - \mathbf{p}_{g,c}^{\mathrm{T}} \mathbf{x}_{gi} \parallel^2 \tag{14}$$

Taking the derivative of the objective function with respect to $\mathbf{p}_{g,c}$, the optimal solution of $\mathbf{p}_{g,c}$ can be obtained:

$$\mathbf{p}_{g,c} = \left( \mathbf{I}_{D \times D} + \sum_{i=1}^{N} \mathbf{x}_{gi} \mathbf{x}_{gi}^{\mathrm{T}} \right)^{-1} \cdot \left( \lambda_{\mathbf{p}_g} \sum_{i=1}^{N} \mathbf{x}_{gi} y_{ic} \right) \tag{15}$$

### 2.5. Multiview TSK Fuzzy System via HSIC

The complementarity principle is an important criterion in multiview learning. In our data, each view corresponds to a group of features, so each view has unique information. Therefore, making accurate predictions requires integrating information from each view. In this paper, we apply the Hilbert–Schmidt independence criterion (HSIC) to realize the idea of the complementarity principle. The HSIC is used to measure the independence between different views. The independence of each view can reduce redundant information and enhance complementarity. According to the method of Cao et al. [19], the empirical version of HSIC is summarized as follows:

$$\mathrm{HSIC} \left( \mathbf{E}^v, \mathbf{E}^h \right) = (n-1)^{-2} \mathrm{Tr} \left( \mathbf{K}^v \mathbf{H} \mathbf{K}^h \mathbf{H} \right) \tag{16}$$

where $\mathbf{E}^v$ is the prediction error in view $v$ and $\mathbf{K}^v$ is the Gram matrix in view $v$. We set $\mathbf{K}^v = \mathbf{E}^v (\mathbf{E}^v)^{\mathrm{T}}$. $h_{ij} = \delta_{ij} - 1/n$ centers the Gram matrix to have a zero mean in the feature space. For notational convenience, we ignore the scaling factor $(n-1)^{-2}$. When all views except view $v$ are fixed, we minimize the following function:

$$\sum_{h=1;h \neq v}^{V} \mathrm{HSIC} \left( \mathbf{E}^v, \mathbf{E}^h \right) = \mathrm{Tr} \left( \sum_{h=1;h \neq v}^{V} \mathbf{H} \mathbf{K}^v \mathbf{H} \mathbf{K}^h \right) \tag{17}$$

$$= \mathrm{Tr} \left( \sum_{h=1;h \neq v}^{V} (\mathbf{E}^v)^{\mathrm{T}} \mathbf{H} \mathbf{K}^h \mathbf{H} \mathbf{E}^v \right) = \mathrm{Tr} \left( (\mathbf{E}^v)^{\mathrm{T}} \mathbf{G}^v \mathbf{E}^v \right) \tag{18}$$

where

$$\mathbf{G}^v = \sum_{h=1;h \neq v}^{V} \mathbf{H} \mathbf{K}^h \mathbf{H} \tag{19}$$



With the HSIC, we obtain the following objective function:

$$\min_{\mathbf{P}^v, \mathbf{E}^v} J_{\text{TSK-HSIC}}(\mathbf{P}^v, \mathbf{E}^v) = \text{Tr}\left(\frac{1}{2}\sum_{v=1}^{V}(\mathbf{P}^v)^{\text{T}}\mathbf{P}^v + \frac{\lambda_{\mathbf{P}}^v}{2}\sum_{v=1}^{V}(\mathbf{E}^v)^{\text{T}}\mathbf{E}^v + \frac{\gamma}{2}\sum_{v=1}^{V}(\mathbf{E}^v)^{\text{T}}\mathbf{G}^v\mathbf{E}^v\right) \quad (20)$$

$$\text{s.t.}\,\mathbf{Y}_{vec} = \mathbf{X}_g^v\mathbf{P}^v + \mathbf{E}^v \text{ for } v = 1, 2, \dots, V \quad (21)$$

where $\mathbf{X}_g^v$ is the matrix of the input data of view $v$ through the transformation of Equation (10), $\mathbf{Y}_{vec}$ is the matrix obtained from the label vector through one-hot coding, $\mathbf{E}^v$ represents the error matrix of view $v$, $\mathbf{Y}_{vec}$, $\mathbf{E}^v \in \mathbf{R}^{N \times C}$, and $\mathbf{P}^v$ is a matrix composed of the consequent parameters of the TSK fuzzy system, $\mathbf{P}^v \in \mathbf{R}^{K(D^v+1) \times C}$. $V$ is the number of all views, $C$ is the total number of classes, $N$ is the total number of data samples, $K$ is the total number of rules, and $D$ is the number of data dimensions. $\gamma, \lambda_{\mathbf{P}}^v$ are regularization parameters. Their values can be obtained by cross validation. The Lagrange function of this problem is defined as:

$$\mathcal{L}(\mathbf{P}^v, \mathbf{E}^v) = J_{\text{TSK-HSIC}}(\mathbf{P}^v, \mathbf{E}^v) - \text{Tr}(\sum_{v=1}^{V}\boldsymbol{\alpha}^{v\text{T}}\left(\mathbf{X}_g^v\mathbf{P}^v + \mathbf{E}^v - \mathbf{Y}_{vec}\right)) \quad (22)$$

Let $\partial\mathcal{L}/\partial\mathbf{P}^v = 0$, $\partial\mathcal{L}/\partial\mathbf{E}^v = 0$ and $\partial\mathcal{L}/\partial\boldsymbol{\alpha}^v = 0$:

$$\begin{cases} \frac{\partial\mathcal{L}}{\partial\mathbf{P}^v} = 0 \rightarrow \mathbf{P}^v = \left(\mathbf{X}_g^v\right)^{\text{T}}\boldsymbol{\alpha}^v \\ \frac{\partial\mathcal{L}}{\partial\mathbf{E}^v} = 0 \rightarrow \lambda_{\mathbf{P}}^v\mathbf{E}^v + \gamma\mathbf{G}^v\mathbf{E}^v = \boldsymbol{\alpha}^v \\ \frac{\partial\mathcal{L}}{\partial\boldsymbol{\alpha}^v} = 0 \rightarrow \mathbf{Y}_{vec} = \mathbf{X}_g^v\mathbf{P}^v + \mathbf{E}^v \\ \quad where\ v = 1, 2, \dots, V \end{cases} \quad (23)$$

solving these equations, the solution of $\mathbf{P}^v$ and $\mathbf{E}^v$ can be obtained

$$\mathbf{E}^v = (\lambda_{\mathbf{P}}^v\mathbf{X}_g^v\left(\mathbf{X}_g^v\right)^{\text{T}} + \gamma\mathbf{X}_g^v\left(\mathbf{X}_g^v\right)^{\text{T}}\mathbf{G}^v + \mathbf{I}_{\text{N}})^{-1}\mathbf{Y}_{vec} \quad (24)$$

$$\mathbf{P}^v = \left(\mathbf{X}_g^v\right)^{T}(\lambda_P^v\mathbf{I}_N + \gamma\mathbf{G}^v)\mathbf{E}^v = \left(\mathbf{X}_g^v\right)^{T}(\lambda_P^v\mathbf{I}_N + \gamma\mathbf{G}^v)(\lambda_P^v\mathbf{X}_g^v\left(\mathbf{X}_g^v\right)^{T} + \gamma\mathbf{X}_g^v\left(\mathbf{X}_g^v\right)^{T}\mathbf{G}^v + \mathbf{I}_N)^{-1}\mathbf{Y}_{vec} \quad (25)$$

### 2.6. Parameter Setting

Selecting appropriate parameters for the model can improve the prediction performance, enhance the generalization ability and avoid overfitting problems. In this study, all parameters were determined for the training set through five-fold cross validation. For the single view TSK fuzzy system, the number of fuzzy rules was taken as the set $\{2, 4, 6, 8, 10\}$, the scaling parameter h in Equation (4) was taken as the set $\{10^{-3}, 10^{-2}, 10^{-1}, 10^0, 10^1, 10^2, 10^3\}$, and the regularization parameter $\lambda_{\mathbf{P}}^v$ was from the set $\{2^{-10}, 2^{-9}, \dots, 2^9, 2^{10}\}$. For the multiview TSK fuzzy system, the regularization parameter $\gamma$ was taken as the set $\{2^{-10}, 2^{-9}, \dots, 2^9, 2^{10}\}$.

### 2.7. Performance Metrics

In this paper, we employed the accuracy (ACC), sensitivity (SN), specificity (SP) and Matthew's correlation coefficient (MCC) to evaluate the performance of our model. The values of ACC, SN and SP are in the range [0, 1], but we want the prediction performance of the proposed model to be higher than the random prediction results, so the acceptable values are [0.5, 1]. Similarly, the range of values for MCC is [−1, 1], and the acceptable values are [0, 1]. Their values are calculated as follows:

$$ACC = \frac{TP + TN}{TP + FP + TN + FN} \quad (26)$$

$$SN = \frac{TP}{TP + FN} \tag{27}$$

$$SP = \frac{TN}{TN + FP} \tag{28}$$

$$MCC = \frac{TP \times TN - FP \times FN}{\sqrt{(TP + FP) \times (TP + FN) \times (TN + FP) \times (TN + FN)}} \tag{29}$$

In these expressions, TP is the number of true positives, TN is the number of true negatives, FP is the number of false positives, and FN is the number of false negatives.

## 3. Results

In this study, we employed two independent datasets, namely, CPP740 and CPP 184. SSA and Pse-AAC describe data from different views. We compared the performance differences between the single features and combined features. Then, the correlation-based feature selection algorithm was employed to remove redundant features. The resulting feature subset was input into the multiview TSK fuzzy system. Finally, cross and independent tests were adopted to validate the empirical performance of the model. The results prove that our method outperformed the state-of-the-art methods in the literature.

### 3.1. Performance Analysis from a Single View

For the CPP740 dataset, we input the SSA and Pse-AAC features into the classic single-view TSK fuzzy system as shown in Equation (14). The results are shown in Table 1. The AAC, SN, SP, MCC of the Pse-AAC feature were 91.1%, 90.5%, 91.6%, and 0.822, respectively, which were better than the SSA feature for all indices. After mixing the two features and taking the multiview approach, it can be seen that the obtained results were not as good as those of a single view, and the ACC, SN, SP, MCC values were 90.1%, 89.2%, 90.8%, and 0.802, respectively. We believe that the outcome changes were not caused by multiview techniques but were worsened by the addition of redundant information and irrelevant features. Therefore, it is essential to employ an appropriate feature selection method.

**Table 1.** Performance of different features on the training set in five-fold cross validation.

| Feature | ACC (%) | SN (%) | SP (%) | MCC |
|---|---|---|---|---|
| Pse-AAC | 91.1 | 90.5 | 91.6 | 0.822 |
| SSA | 88.4 | 86.2 | 90.8 | 0.768 |
| Pse-AAC+SSA | 90.1 | 89.2 | 90.8 | 0.802 |

### 3.2. Performance Analysis after Feature Selection

In this study, the correlation-based feature selection algorithm was utilized because of its excellent performance. We splice the 121-D SSA features with the 50-D Pse-AAC features. The obtained 171-D hybrid features are employed as the input to the feature selection algorithm. Then a 37-dimensional feature subset was obtained, which included 15 SSA features and 22 Pse-AAC features. Among the 121-D SSA features, only 15 features were selected into the optimal feature subset, while 22 of 50-D Pse-AAC were selected. Combined with the performance analysis of the single view, we believe that the Pse-AAC descriptor provides more valuable information for predicting CPPs.

Table 2 shows the five-fold cross validation results after feature selection. It can be observed that the average ACC is 92.2%, the average SN is 90.8%, the average SP is 93.5%, and the average MCC is 0.844. These metrics indicate that there is improvement relative to the prior feature selection, along with a reduction in feature dimensions. CFS improves the computational efficiency as well as the prediction performance of the model.

**Table 2.** Five-fold cross validation of the CPP740 dataset.

| Fold Set | ACC (%) | SN (%) | SP (%) | MCC |
|---|---|---|---|---|
| 1 | 90.5 | 88.2 | 93.1 | 0.812 |
| 2 | 90.5 | 90.8 | 90.3 | 0.811 |
| 3 | 92.6 | 88.2 | 96.3 | 0.852 |
| 4 | 94.0 | 93.4 | 94.4 | 0.878 |
| 5 | 93.2 | 93.2 | 93.2 | 0.865 |
| Average | 92.2 | 90.8 | 93.5 | 0.844 |

### 3.3. Comparative Analysis with Other Classifiers

Table 3 shows the five-fold cross validation results using MV-TSK-FS-HSIC and some classic algorithms with the selected features of the CPP740 dataset. The classic methods included XGBoost, naïve Bayes (NB), and random forest (RF). Among them, the results of RF were the best, with an ACC of 90.4%, SN of 88.6%, SP of 92.2%, MCC of 0.809 and AUC of 0.967. Figure 2 shows the receiver operating characteristic curves of different classifiers for the CPP740 dataset. The AUC of the proposed method is 0.975, which is higher than that of other classic algorithms. Unlike the multiview TSK fuzzy system, we directly input the hybrid features into the classic models. This ignores the relationship between different views and the statistical characteristics of data, so the performance is inferior to that of MV-TSK-FS-HSIC.

**Table 3.** The performance of different classifiers for the CPP740 dataset after feature selection (five-fold cross validation).

| Method | ACC (%) | SN (%) | SP (%) | MCC |
|---|---|---|---|---|
| NB | 90.1 | 85.4 | 94.9 | 0.806 |
| XGBoost | 90.3 | 90.0 | 90.5 | 0.805 |
| RF | 90.4 | 88.6 | 92.2 | 0.809 |
| MV-TSK-FS-HSIC | 92.2 | 90.8 | 93.5 | 0.844 |

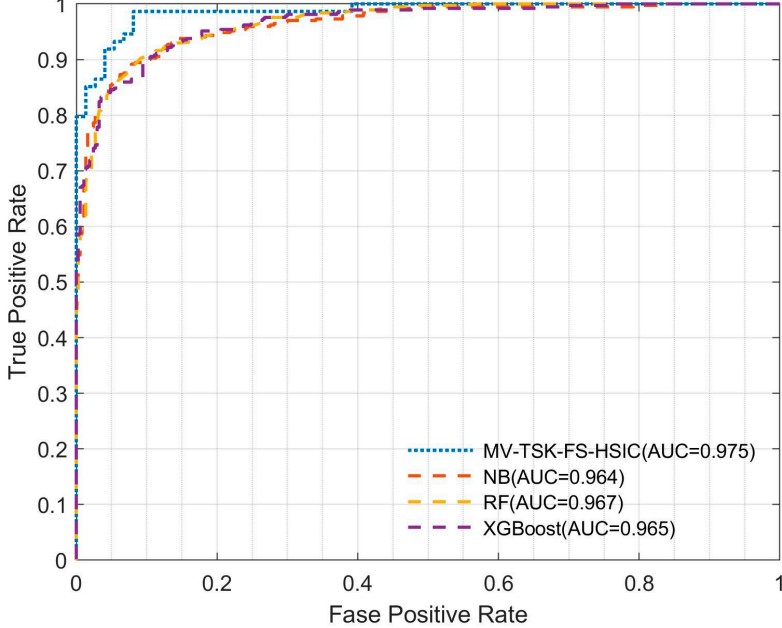

**Figure 2.** Receiver operating characteristic curves of different classifiers after feature selection over five-fold cross validation of the CPP740 dataset.

### 3.4. Comparison Analysis for the CPP740 Dataset

As shown in Table 4, several extant methods, including PPTPP [11], ITP-PRED [12], and PEPred [10] were compared with our method for the CPP740 dataset. The best results of the previous methods were obtained by PEPred. The ACC, SN, SP, MCC and AUC were 91.2%, 90.3%, 92.2%, 0.824, and 0.972, respectively. Compared with PEPred, our method increased the ACC, SN, SP, MCC, and AUC values by 0.01, 0.005, 0.013, 0.02, and 0.003, respectively.

**Table 4.** Comparison of existing methods using the CPP740 dataset and five-fold cross validation.

| Method | ACC (%) | SN (%) | SP (%) | MCC | AUC |
|---|---|---|---|---|---|
| PPTPP | 74.9 | 71.6 | 78.1 | 0.498 | 0.824 |
| ITP-PRED | 89.0 | 86.3 | 93.2 | 0.787 | 0.962 |
| PEPred | 91.2 | 90.3 | 92.2 | 0.824 | 0.972 |
| MV-TSK-FS-HSIC | 92.2 | 90.8 | 93.5 | 0.844 | 0.975 |

### 3.5. Comparison Analysis of an Independent Test Set

To verify the generalization ability of our model, we employed an independent test set with 184 samples. The dataset contains 92 CPPs and the same number of non-CPPs, and there was no overlap with the samples of the training set. The experimental results of the five-fold cross validation are shown in Table 5. The ACC, SN, SP, MCC, and AUC of MV-TSK-FS-HSIC values were 96.2%, 96.7%, 95.7%, 0.924, and 0.990, respectively. Compared with ITP-PRED, our method improved the ACC, SN, MCC, and AUC values by 0.011, 0.039, 0.02, and 0.011, respectively. Only SP was inferior to ITP-PRED. The results prove that our method is superior to the state-of-the-art predictors.

**Table 5.** Comparison of existing methods with an independent test set and five-fold cross validation.

| Method | ACC (%) | SN (%) | SP (%) | MCC | AUC |
|---|---|---|---|---|---|
| PEPred | - | - | - | - | 0.952 |
| PPTPP | - | - | - | - | 0.967 |
| ITP-PRED | 95.1 | 92.8 | 97.8 | 0.904 | 0.989 |
| MV-TSK-FS-HSIC | 96.2 | 96.7 | 95.7 | 0.924 | 0.990 |

## 4. Discussion and Conclusions

In this study, a novel multiview TSK fuzzy system is proposed. First, SSA and Pse-AAC descriptors were employed to convert the original sequence into multiview data. Second, we utilized the correlation-based feature selection algorithm to obtain the optimal feature subset. Finally, the resulting feature subset was input into the multiview TSK fuzzy system based on the HSIC, and a multiview decision result was then obtained. We compared the performance of the proposed model with several classical machine learning algorithms by a five-fold cross-validation. The empirical results demonstrate that the proposed method outperforms the classical methods in terms of ACC, SN, SP, and MCC metrics. We validated the performance of the model using the CPP dataset, the AUC reached 0.975 and 0.990 and the ACC achieved 92.2% and 96.2% on the training and test sets, respectively. The results prove that our method is superior to the existing CPP predictors.

Through feature analysis, we found that Pse-AAC features played a more important role than SSA features in identifying CPPs. We believe that this is because Pse-AAC features contain physicochemical information that can distinguish CPPs from non-CPPs. Also feature selection is necessary to alleviate the overfitting problem, remove redundant features, reduce data dimension and lower computational cost. When the data matrix $\mathbf{X}$ is converted to the input matrix $\mathbf{X}_g$ of the TSK fuzzy system, the dimension increases dramatically, which can seriously deteriorate the computational efficiency of the model if no feature selection is performed.

Although the proposed model has been proven to be effective in experiments, there is still room for improvement. In future research, we expect to introduce a novel multi-view mechanism to investigate the relationships between different views and enhance the interpretability of TSK fuzzy systems.

**Author Contributions:** Conceptualization, Y.D.; methodology, Y.D.; software, P.L.; validation, S.Z.; formal analysis, S.Z. and P.L.; investigation, S.Z.; resources, Q.Z.; data curation, P.L.; writing—original draft preparation, P.L.; writing—review and editing, Y.D.; visualization, P.L.; supervision, Q.Z.; project administration, Q.Z. All authors have read and agreed to the published version of the manuscript.

**Funding:** The work was supported by the National Natural Science Foundation of China (No. 62172076, 61902271), and the Municipal Government of Quzhou under Grant Number 2020D003 and 2021D004.

**Institutional Review Board Statement:** Not applicable.

**Informed Consent Statement:** Not applicable.

**Data Availability Statement:** The data used in this paper can be downloaded at http://lab.malab.cn/~acy/BioseqData/Protein/ITP-Pred.rar (accessed on 16 January 2022).

**Acknowledgments:** We thank Abd El-Latif Hesham for his contribution to this paper.

**Conflicts of Interest:** The authors declare no conflict of interest.

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
