# Peer review of "Prediction of Cell-Penetrating Peptides Using a Novel HSIC-Based Multiview TSK Fuzzy System"

_applsci, doi:10.3390/app12115383_

Round 1

Reviewer 1 Report

The language of this manuscript should be improved and simplified. Besides, there are some major concerns:

  1. Table 2 is designed to demonstrate that Pse-AAC descriptor provides more valuable information for predicting CPPs. To this end, I suggest checking the performance of using some random subsets (also 37 features) of features in this task.
  2. Table 3 looks like being partially covered by Figure 2.
  3. Figure 3 and Table 4 are just redundant, please choose one to retain.
  4. Please briefly introduce the "independent test set" used in Table 5.

Reviewer 2 Report

Comments (General)

  • The keywords shall have to be five. The authors should add some keywords.
  • The authors shall have to present the specific research problem in “Abstract” first.
  • The authors shall have to check all equations in details and some parameters are missing.
  • The authors shall have to present the system’s robustness, recommendations on proposed system, and limitations of the proposed system.
  • The conclusion section shall have to be modified based on the outcomes of the proposed system and the authors shall have to mention some significant points with numerical values.

Comments (Specific)

  • The authors shall have to mention the accuracy and performance percentage of the proposed system in numerical value in “Abstract”.
  • The authors shall have to express the mathematical model of the similarity loss function to optimize the model parameters through backpropagation.
  • The authors shall have to illustrate the feature table for feature selection.
  • In performance metric, the authors shall have to declare the accepted percentage range for the proposed system and it could be provided to compare the performance values from the applications.
  • The authors shall have to recheck the equations 12 and 13.
  • Equation (14.4) shall have to be checked again. The mathematical model is strange one and the authors shall have to confirm that equation for publication.
  • The authors shall have to calculate the rule table with TP, TN, FP, FN and so on. It is very important for the confirmation of the proposed system’s exactness.
  • The authors shall have to verify the receiver operating characteristic curves with true negative rate vs false negative rate also.
  • The authors had declared two datasets for analyses and the authors shall have to verify the results of the proposed system with those two datasets.
  • The authors shall have to point-out the significant points with numerical values in statistics table for the verification of the proposed system. (Before the last section of Discussions and Conclusion).

Round 2

Reviewer 1 Report

I agree to accept